# Major Complication Following Kawasaki Disease in an Infant—The Development of Apical Infarction and Aneurysm Formation

**DOI:** 10.3390/children8110981

**Published:** 2021-10-29

**Authors:** Samuel Menahem, Jeffrey Lefkovits

**Affiliations:** 1Melbourne Children’s Cardiology/Adult Congenital Heart, Melbourne, VIC 3161, Australia; 2Department of Paediatrics, Monash University, Melbourne, VIC 3800, Australia; 3Department of Paediatrics, University of Melbourne, Melbourne, VIC 3010, Australia; 4Department of Cardiology, Royal Melbourne Hospital, Melbourne, VIC 3050, Australia; lefkovits@me.com; 5Department of Epidemiology and Preventive Medicine, Monash University, Melbourne, VIC 3800, Australia

**Keywords:** Kawasaki disease, myocardial infarction, ventricular apical aneurysm, coronary artery calcification

## Abstract

Considerable advances have occurred in the understanding of Kawasaki disease, with a substantial drop in morbidity and mortality following the infusion of gamma globulin during the acute phase. Nevertheless, major complications may still occur. A 27-year-old male presented as an infant of 11 weeks when he was diagnosed as having Kawasaki disease. He was appropriately treated with aspirin and a gamma globulin infusion following his diagnosis 5 days after the onset of his illness. Despite that, he went on to develop coronary aneurysms. He represented a few weeks later with a history of inconsolable crying associated with pallor, suggestive of ischaemic chest pain. A repeat echocardiogram revealed infarction of the apex of the left ventricle with localised thrombus formation. There were persistent aneurysms within both coronary artery systems. A further infusion of gamma globulin was given. In view of the thrombus formation, he was started on warfarin. The thrombus gradually resolved with the development of a clearly defined left ventricular apical aneurysm. He has remained on warfarin, aiming for an international normalised ratio (INR) level of 2 to 2.5. He developed mild left ventricular dysfunction during late childhood, which improved following the commencement of an angiotensin-converting enzyme (ACE) inhibitor. Despite his ventricular aneurysm, there has been no documented evidence of ventricular tachycardia over the years. Repeated testing initially by nuclear perfusion scans and then by stress echocardiograms failed to show any inducible ischaemia apart from the apical ventricular aneurysm. A recent computed tomography (CT) coronary angiogram revealed an ectatic origin of the left main and the right coronary arteries with mild calcification involving the mid-portion of the latter and slight calcification of the former. His raised cholesterol level has responded well to a statin. Despite the persistence of the ventricular aneurysm, he continues to be managed conservatively, as he has remained well. The question arises as to what the long-term implications are of his left ventricle apical aneurysm. Should it be excised? Is he at risk for ventricular tachycardia and sudden death? In addition, although the coronary aneurysms have resolved, the CT coronary angiogram shows calcium plaques in both coronary arteries at the site of the earlier aneurysms. This finding raises the question as to whether all children who develop coronary artery aneurysms following Kawasaki disease should have a CT coronary angiogram performed in adulthood.

## 1. Introduction

Considerable advances have occurred in the understanding of Kawasaki disease—its pathogenesis, clinical manifestations, its short- and long-term complications and the best management strategies [1,2]. While its aetiology remains elusive, it is now recognised as occurring worldwide, though it remains most common among Japanese children with a rate of 264/100,000, with white children having the lowest incidence, 13.7/100,000 [2]. The first case diagnosed in Australia, though retrospectively, and only after a visit from a Japanese paediatrician who described the then relatively new condition diagnosed by Dr Kawasaki—previously called mucocutaneous lymph node disease—died suddenly in the third week of his illness [3]. The efficacy of a gamma globulin infusion has greatly altered the natural history of the condition, substantially reducing the incidence of coronary artery involvement and aneurysm formation from 24% to 4% [4], with a resultant reduction in thrombus formation and myocardial ischaemia, with all their consequences. That, in turn, has significantly improved the prognosis of those affected patients reducing their morbidity and resultant mortality.

Despite these advances, we describe an infant who, in spite of receiving two infusions of gamma globulin early in the course of his illness, and being maintained on a low dose of aspirin, still went on to develop coronary artery involvement, and sustained a myocardial infarct with the subsequent development of a left ventricular apical aneurysm with a somewhat unclear and guarded future.

## 2. Case Report

“Bob” was admitted at the age of 11 weeks, febrile and very irritable. He had presented to the local hospital 5 days earlier with a history of being unduly quiet, lethargic and febrile, with grunting respirations. He went on to develop bilateral conjunctivitis, lymphadenopathy and an evolving generalised rash. An initial chest X-ray and lumbar function were clear. He then developed palmar erythema and red swollen lips. His white cell count was raised and well as his erythrocyte sedimentation rate (ESR). When seen by the local paediatrician, Kawasaki disease was diagnosed. He was given an infusion of gamma globulin 2 gm/kgl and commenced on high dose aspirin.

Little improvement occurred over the next few days. Transfer to the Royal Children’s Hospital was arranged. On arrival, his findings were as described. Apart from an innocent murmur, the rest of his cardiac examination was normal. His electrocardiogram (ECG) was within the normal limits, while his echocardiogram was normal. His white cell count and ESR were raised, he had a mild normocytic anaemia and a thrombocytosis. He improved over the next week and was discharged home. His subsequent course was variable, with intermittent fever, irritability and a fluctuating rash. He was re-admitted to the local hospital only to be transferred back to Melbourne a few days later. He then appeared rather unwell, pale with a generalised rash, hepatosplenomegaly, cervical lymphadenopathy and early desquamation of the tips of his fingers and toes. A repeat echocardiogram now showed diffuse dilatation of both coronary arteries, with an aneurysm involving the right coronary artery. A further dose of intravenous gamma globulin 2 g/kg was given, while his aspirin dosage increased. His mild anaemia improved. There was a fall in his ESR and platelet count. His rash came and went. His liver and splenic enlargement settled with a return to normal of his liver enzymes.

Once home, his irritability and fever fluctuated. A few days later, he developed a distressing cough for which he was readmitted to the local hospital for observation. A further chest X-ray was clear. He was discharged a few days later. His cough persisted, although it did improve. He still, however, showed a fluctuating fever, rash and varying irritability. On day 49 of his illness, mother noted that he had become exceedingly irritable. He went pale, cold and clammy on crying. The possibility of myocardial ischaemia/infarction was considered by his paediatrician. An ECG showed ischaemic changes. The infant was then flown back to Melbourne. His clinical findings were as described above. His echocardiogram showed an infarct of the apex of the left ventricle with resultant paradoxical movement and the presence of a laminated thrombus (Figure 1a). Intravenous heparin was commenced, subsequently changing over to warfarin, aiming for an INR level of 2–2.5 prior to his discharge [5].

Over the subsequent months, his thrombus gradually resolved, but he was left with the ventricular apical aneurysm which went on to become thin-walled, with clear evidence of paradoxical movement, dilating on left ventricular contraction (see Figure 1b). His overall left ventricular function, however, remained good.

At the age of 9 months, he had a selective coronary angiogram performed. That showed a 7 mm aneurysm in the left anterior descending coronary artery just distal to its bifurcation from the left main coronary artery. In addition, he had a 10 mm aneurysm involving the proximal right coronary artery.

Over the years, “Bob” has remained well, remaining on warfarin with no recurrence of his apical thrombus. There has been no documented history of an arrhythmia, while repeated 24 h Holter monitoring has been normal. Despite a normal exercise tolerance, he went on to develop, at the age of 9 years, echocardiographic evidence of mild LV dilatation with reduced contractility. He was started on an ACE inhibitor and has remained on 5 mg of lisinopril daily with a return to normal left ventricular function and maintenance of a normal exercise tolerance.

Despite being symptom-free, active measures were undertaken to ensure that he was not developing any evidence of further coronary artery ischaemia as a result of the narrowing of the affected coronary arteries. Following the visit of an expert from Japan, biennial exercise nuclear Sestamibi studies were carried out [2,6]. These regular assessments did not show any areas of inducible myocardial ischaemic, except for the persistent apical defect arising from his earlier infarct. Good left ventricular function was maintained with a normal ejection fraction, apart from the apical hypokinesis. Nuclear scans have since been replaced by yearly exercise stress echocardiograms [7,8], which to date have not shown the development of other areas of inducible ischaemia, apart from the known apical infarct. Bone density estimations have remained normal despite his long-term warfarin intake.

At the age of 25 years, “Bob” had a CT coronary angiogram [9]. The reconstituted CT image clearly demonstrated the pale thin walled apical aneurysm (Figure 1c,d). The angiogram showed mild stenosis of the proximal to mid left anterior descending artery with a mildly ectatic (up to 4.5 mm diameter) mid portion and a mixed plaque in the mid-vessel resulting in mild stenosis (Figure 2). There was also mild ectasia of the proximal right coronary artery (up to 4.6 mm in diameter) with a dense calcified plaque in its mid portion (Figure 3). The modified calcium score was 418, which was above the 95th percentile for age and sex [10]. Further surveillance was recommended. However, at the time, he was noted to have an elevated total cholesterol level of 5.9 mmol/L with a raised low-density lipoprotein of 3.9 mmol/L. He was started on 10 mg rosuvastatin [11], with a drop in his total cholesterol level to 3.7 mmol/L.

## 3. Discussion

Despite what would appear at the time to have been optimal treatment, having commenced with a high dose of aspirin followed by one and then a second infusion of gamma globulin, the first following his diagnosis 5 days after the onset of his illness, and the second when he presented almost 2 weeks later still unwell, “Bob” went on to develop myocardial ischaemia, myocardial infarction and an apical aneurysm, despite reducing the aspirin dose to achieve an antiplatelet effect [12,13]. Despite his initial gamma globulin infusion, when he was initially readmitted almost 2 weeks later, abnormalities of the coronary arteries including an aneurysm of the right coronary artery were noted on a repeat echocardiogram. Such abnormalities have been observed in up to 20% of individuals, even after receiving a gamma globulin infusion, though only 4% to 5% still seem to develop coronary artery aneurysms [2,14]. The development of a laminated thrombus warranted warfarinisation [5]. He has remained free of thrombus since. Importantly, the paediatrician involved in his care noted that when he cried following his second discharge from hospital, he went pale and somewhat limp, which would suggest a more sinister cause rather than the usual lusty crying that babies not infrequently exhibit. That prompted the early referral of “Bob” back to the tertiary hospital. Was his readmission because of a recurrence of his Kawasaki disease which appeared to have only partially settled after his first gamma globulin infusion [15], or was it more likely a continuation of his initial and only episode of Kawasaki disease, improving but not fully settling after his initial treatment which resulted in the complications described [2]?

It is a rare event to sustain a myocardial infarction in the acute phase of Kawasaki disease. Manlhiot et al. [16] suggested that it occurs in 1% of those with an aneurysm that had a Z score of less than 10 and an absolute dimension of less than 8 mm. Neither feature was observed with “Bob”. The acute ischaemia and/or infarction was more likely related to a thrombus involving the acutely inflamed coronary arteries despite his low dose of aspirin, rather than due to increasing stenosis, which may take many years to develop, particularly more so if there is a giant aneurysm [12,17]. Such an event may have explained the sudden demise of our first patient in the third week of his illness [3]. Kato [18] provided a clear description of the progression of the coronary artery changes in Kawasaki disease, progressing from a normal coronary to one of mild dilatation which may be transient over 4 to 6 weeks, or following necrotizing arteritis may result in aneurysm formation with destruction of the intima, elastica externa, media, and variably the adventitia. This may lead to a subacute/chronic vasculitis with luminal myofibroblastic proliferation and laminar non-occlusive thrombosis. There may be a return to normal luminal dimension or further progression and possible interaction with atherosclerosis risk factors, resulting in calcification and complex stenosis, as is now being observed in our patient as he reaches adulthood. Alternatively, the coronary aneurysms may progress to occlusive thrombosis formation, leading to a myocardial infarction with subsequent organisation of the thrombus and recanalization.

The advent of stress echocardiography has generally done away with the need for repeated nuclear scans which involves some radiation, to determine the presence or otherwise of inducible myocardial ischaemia [7]. However, there is still the need from time to time to review the coronary arteries. The CT coronary angiogram has become a relatively non-invasive but meaningful investigation [9]. “Bob” has demonstrated an increased likelihood of developing ischaemic heart disease in adult life following his episode of Kawasaki disease as an infant [18,19,20]. How active should one be with respect to its treatment remains unclear, though the importance that he leads a good lifestyle has been repeatedly emphasised [2]. He remains active, is normotensive, he does not smoke, and has a normal BMI. His cholesterol levels were raised, and he is currently on appropriate treatment. His CT coronary findings, silent to date despite active exercise testing, suggests the wisdom of carrying out such an investigation in all young adults who had previously developed coronary aneurysms from Kawasaki disease. The involvement of an adult interventionalist cardiologist becomes essential in the long-term follow-up once coronary abnormalities are noted.

There remains the question as to how best to manage “Bob’s” apical aneurysm. Ventricular aneurysms are rare in childhood. A case report [21] described a 4-year-old who developed a thin-walled left ventricular apical aneurysm in the course of his treatment for acute lymphoblastic leukaemia. It was successfully excised because of the fear of possible rupture. Ventricular aneurysms may be congenital and ventricular arrhythmias have been described as occurring in adulthood [22]. Resection of a congenital left ventricular aneurysm was successfully carried out in a girl aged 11 with associated mitral regurgitation. She improved following resection of the aneurysm in addition to a mitral valve annuloplasty [23]. A further case report described the successful resection of a left ventricular aneurysm with an intramural thrombus in a toddler with combined immunodeficiency syndrome [24]. There do not appear to be previous reports of the management of a left ventricular aneurysm arising from a myocardial infarction in childhood, though theoretically that possibility was raised as a potential cause. “Bob” has had neither a history of ventricular arrhythmias nor was it likely that his aneurysm would rupture. His ventricular function improved and has been stable while being maintained on an ACE inhibitor. There has been no recurrence of thrombus in the aneurysm since he was started on warfarin. The absence of the above indications did away with the need for surgical intervention and/or antiarrhythmic medication during childhood. As he has reached adulthood, experience arising from studies where a ventricular aneurysm has occurred from myocardial ischaemia becomes increasingly relevant. Following myocardial infarction, resection of the aneurysm is generally not considered unless there is ongoing evidence of poor left ventricular function, refractory ventricular arrhythmias or recurrent thromboembolism despite appropriate anticoagulant therapy [25,26,27]. Surgical resection may well improve function [28]. “Bob” responded well to an ACE inhibitor. The additional question regards his risk of developing ventricular arrhythmias. To date, this has not been documented, and no prophylactic anti-arrhythmics have been prescribed [22,29], meaning that he continues to be managed conservatively.

## 4. Conclusions

Kawasaki disease, with its emphasis on early diagnosis and improved management strategies, generally has a good short- and long-term outlook. Nevertheless, serious complications may occur despite optimal treatment, as evidenced by our infant who sustained a myocardial infarct and went on to develop a left ventricular apical aneurysm. Conservative management appears to be reasonable at this stage, though he shows evidence of abnormal coronaries on a CT angiogram. He, as with all patients who develop coronary artery involvement in Kawasaki disease, warrants ongoing surveillance and expert management if and when required.

## Figures and Tables

**Figure 1 children-08-00981-f001:**
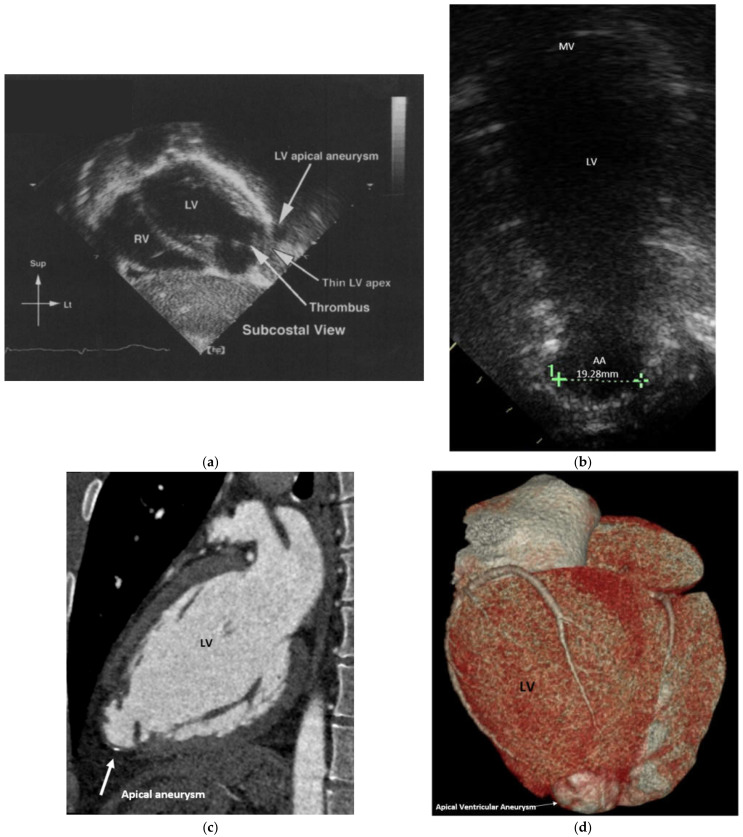
(**a**) Apical aneurysm of the left ventricle. Note the thin wall of the aneurysm and the development of a laminated thrombus. Patient aged 11 months. Reprinted with permission from Cambridge University Press. Licence number 5156740066045 [5]; (**b**) current two-chamber apical view of left ventricle showing the apical aneurysm; (**c**) recent CT image of the left ventricle showing the apical aneurysm; (**d**) reconstituted CT image of the posterior aspect of the left ventricle showing the pale ischaemic apical aneurysm. Sup = superior, Lt = left, LV = left ventricle, RV = right ventricle, MV = mitral valve, AA = apical aneurysm.

**Figure 2 children-08-00981-f002:**
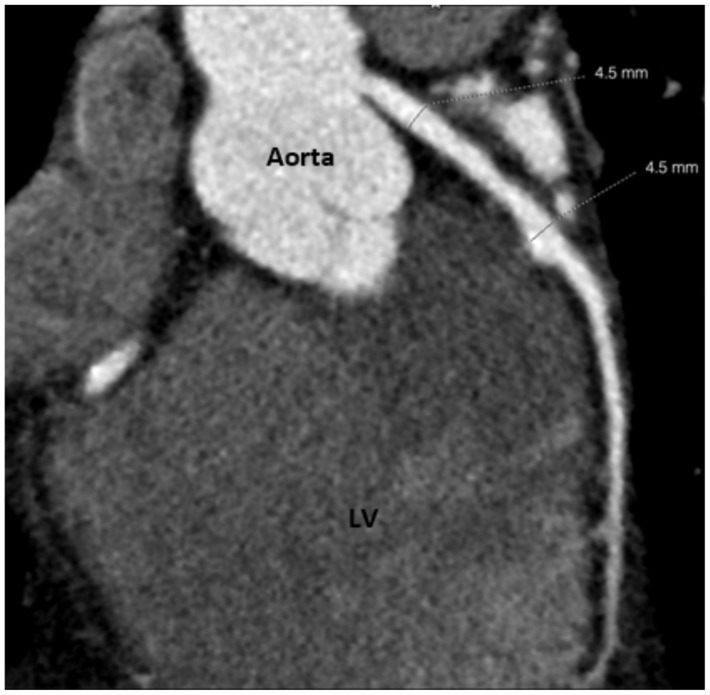
CT angiogram of left anterior descending artery showing persistent mild dilatation of its mid portion and a calcium plague. LV = left ventricle.

**Figure 3 children-08-00981-f003:**
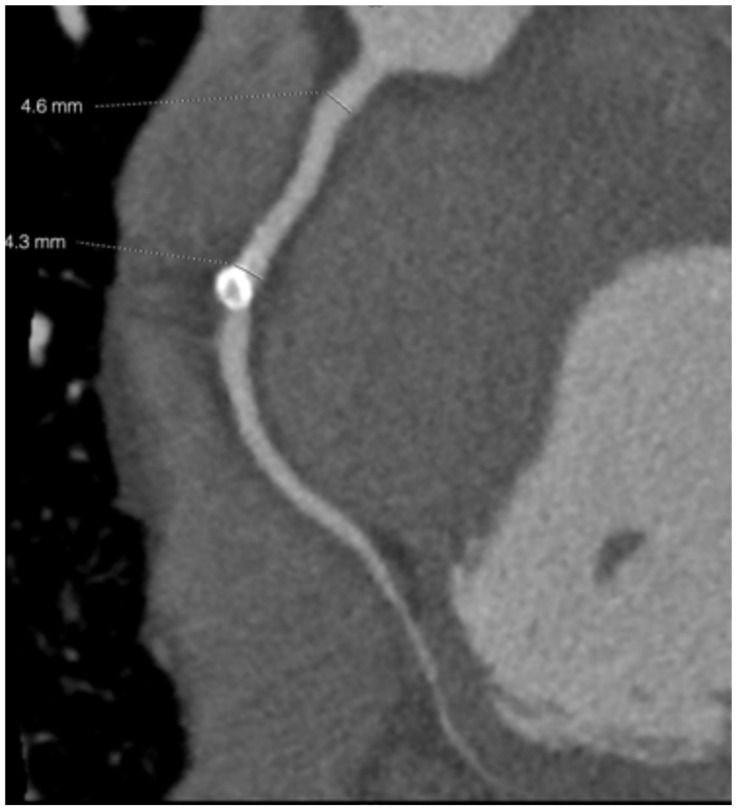
CT angiogram of right coronary artery with dense calcification of its mid portion.

## Data Availability

Data supporting details may be found at the corresponding author’s address at 53 Kooyong Road, Caulfield North VIC 3161, Australia.

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
