# Peer review of "Major Complication Following Kawasaki Disease in an Infant—The Development of Apical Infarction and Aneurysm Formation"

_children, 2021, doi:10.3390/children8110981_

Round 1

Reviewer 1 Report

Overall: The authors present an interesting case that I think can help guide other patient care, however will require some rewording to improve impact. For example, I recommend reducing some details of the case to allow more elaborate discussion with more comparisons about ventricular aneurysm management in other scenarios, ideally in children. 

Here are my comments by section: 

Abstract: 

The reference to "full hand suggestive of Kawasaki's" is confusing to me. While there can be extremity changes in KD, KD is clinical diagnosis that requires other criteria as well. Recommend saying something like, "A 27 year old male presented as an infant at the age of 3 months when he was diagnosed with  Kawasaki disease." 

Case report: The timing of IVIG infusion is important to note as complication risk differs whether the IVIG was infused in the first 10 days or not. 

There are details that can be eliminated that do not add to the message the authors are trying to convey. For example, the bone scan info doesn't add to the story. However, there are also details missing that are important, such as the timing of the first IVIG infusion since that correlates with complications following KD. 

Figures: recommend removing the date that can be seen in upper right corner of figure 1a for HIPAA purposes 

Discussion: 

line 136, the is an "a" before ventricular arrhythmias that should be removed. 

I think it is important to note that in infants, coronary artery abdnormalities are not uncommon and can be present in ~18% of infants despite treatment. As this likely led to infarction. It would be interesting and helpful if the authors noted how common or uncommon infarctions are in children following KD.

The authors note ventricular aneurysm management in adults but further expanding on this and how even in a child watchful waiting as appeared to work well. Noting other literature of ventricular aneurysms and management in children would also be helpful. This literature may be present in CT surgery or congenital heart disease related articles. Even if no such literature is present, noting that would be helpful. 

Author Response

  1. The section on the management of ventricular aneurysm in children has been expanded.
  2. The Introduction has been re-written and expanded. However we decided not to provide an overview of the treatment of Kawasaki disease as suggested by your colleague, as I am sure other papers within the issue will deal with that. We have concentrated specifically on the issues that were pertinent to our case report.
  3. The relevant sentence has been changed accordingly.
  4. The timing of the IVIG infusion has now been stated
  5. The line related to the bone scan has been re-written but has still been included as that is an ongoing issue for children who are on long term Warfarin.
  6. The date has been removed from Figure 1a.
  7. Discussion. “a” has been removed before “ventricular arrhythmias”.
  8. The comment relating to coronary artery abnormalities has now been included in the body of the paper.
  9. A further section has been added with respect to ventricular aneurysm management in a child. However there is very little literature available with respect to the management of a ventricular aneurysm arising from an ischaemic cause. As suggested congenital heart and other literature have been reviewed.

Reviewer 2 Report

The authors present a case report of a 27-year-old male patient who developed Kawasaki Disease (KD) at the age of 3 months, was managed appropriately, but still developed coronary artery aneurysms and apical infarction. At present, the patient suffers from mild left ventricular dysfunction and left ventricle apical aneurysm. I thank the authors for their work and have some comments to improve their manuscript:

Major Comments:

  1. Lines 34-35: please elaborate for the reader what the “recommended treatment” is in KD. Ideally, please accompany this with the most recent guidelines.
  2. Lines 38-39: please elaborate on what clinical features were present that suggested the diagnosis of KD to best inform readers in practice.
  3. The report would benefit from a discussion of surgical indications, i.e., at what stage should a patient with KD who develops coronary artery aneurysms be operated on according to the most recent evidence and/or guidelines?
  4. The report would also benefit from a brief discussion of epidemiological factors: how common is KD? How common are coronary artery aneurysms with and without treatment? Apical aneurysm? Ventricular arrhythmias?

Minor Comment:

  1. Abbreviations should be written in full the first time before being used (e.g., Line 24: ACE, Line 25: CT, Line 47: ECG, Line 51: INR, etc.).

Author Response

  1. The Introduction has been rewritten and the recommended treatment has been stated.
  2. The clinical factors have been added.
  3. We have briefly dealt with the timing for surgery if any following the development of stenosis associated with a coronary aneurysm especially if great. It is very likely a more detailed response will probably be provided in other submissions.
  4. The additional epidemiological information requested has been added and the frequency stated if such information is available.

Minor Comments – The abbreviations have been dealt with.

Round 2

Reviewer 1 Report

Prior recommendations were addressed and now tells a compelling story for readers in regards to potential complications later in life following KD and infarction related to KD in children. 

I'd still be interested to know if IVIG was infused within 10 days of symptom onset for KD. A reader has to assume that was the case as the authors say "appropriately" and "timely." If the timing is known may be worth specifying so readers aren't guessing. I suspect pediatricians would be interested in knowing. 

Author Response

I have altered the paper accordingly as suggested by Reviewer 1 to state the time involved between the onset of his illness and the first infusion of gamma globulin. I have been fortunate subsequent to speaking to mother, that she had copies of the summaries of the inpatient stays of JB, as his notes are no longer available at the hospital. That has resulted in me re-writing the case report, providing clearer details of JB’s earlier history.

Reviewer 2 Report

The authors have adequately addressed and incorporated all comments. I thank the authors and applaud them for their work.

Author Response

Thanks for your kind comments.